Bioactive marine metabolites derived from the Persian Gulf compared to the Red Sea: similar environments and wide gap in drug discovery

Abuhijjleh Reham K. 1
Shabbir Samiullah 1
Al-Abd Ahmed M. 1 2 ahmedmalabd@pharma.asu.edu.eg
Jiaan Nada H. 1
Alshamil Shahad 1
El-labbad Eman M. 1 3
Khalifa Sherief I. 1
1 Department of Pharmaceutical Sciences, College of Pharmacy, Gulf Medical University , Ajman , United Arab Emirates
2 Pharmacology Department, Medical Division, National Research Centre , Giza, Cairo , Egypt
3 Pharmaceutical Chemistry Department, Faculty of Pharmacy, Ain Shams University , Cairo , Egypt
Lenardao Eder
Electronic publication date: 2021 Jul 28
Publication date: 2021
Volume: 9
Electronic Location ID: e11778
Received 2021 Mar 8; Accepted 2021 Jun 24
Copyright: © 2021 Abuhijjleh et al.
Copyright year: 2021
Copyright holder: Abuhijjleh et al.
License: This is an open access article distributed under the terms of the Creative Commons Attribution License, which permits unrestricted use, distribution, reproduction and adaptation in any medium and for any purpose provided that it is properly attributed. For attribution, the original author(s), title, publication source (PeerJ) and either DOI or URL of the article must be cited.
License URL: https://creativecommons.org/licenses/by/4.0/

Keywords: Marine metabolites, Red sea, Gulf of Oman, Fauna, Aspergillus sp., Persian Gulf, Arabian Gulf

Funding: The authors received no funding for this work.

==============================
Marine life has provided mankind with unique and extraordinary chemical structures and scaffolds with potent biological activities. Many organisms and secondary metabolites derived from fungi and symbionts are found to be more environmentally friendly to study than the marine corals per se. Marine symbionts such as Aspergillus sp., a fungus, which can be isolated and grown in the lab would be a potential and continuous source of bioactive natural compounds without affecting the marine environment. The Red Sea is known for its biodiversity and is well-studied in terms of its marine-derived bioactive metabolites. The harsh environmental conditions lead to the development of unique metabolic pathways. This, in turn, results in enhanced synthesis and release of toxic and bioactive chemicals. Interestingly, the Persian Gulf and the Gulf of Oman carry a variety of environmental stresses, some of which are similar to the Red Sea. When compared to the Red Sea, the Persian Gulf has been shown to be rich in marine fungi as well, and is, therefore, expected to contain elaborate and interesting bioactive compounds. Such compounds may or may not be similar to the ones isolated from the Red Sea environment. Astoundingly, there are a very limited number of studies on the bioactive portfolio of marine-derived metabolites from the Persian Gulf and the Gulf of Oman. In this perspective, we are looking at the Red Sea as a comparator marine environment and bioactive materials repertoire to provide a futuristic perspective on the potential of the understudied and possibly overlooked bioactive metabolites derived from the marine life of the Persian Gulf and the Gulf of Oman despite its proven biodiversity and harsher environmental stress.

Introduction

Over the past couple of decades, the number of totally synthetic FDA approved drugs is significantly decreasing (Patridge et al., 2016). Yet, naturally occurring compounds with unique and new scaffold characteristics constitute a virgin area of drug discovery and an inspiring space for semisynthetic and synthetic drug discovery research. In the duration 1980–2010, 26% of the newly approved drugs (1,355 drugs) were natural products or natural compounds’ derivatives (Newman & Cragg, 2012). Terrestrial plant-derived natural products have a long history in supplying mankind with a giant library of unique stereo-chemically complex compounds with diverse biological activities. On the other hand, marine life provided drug discovery researchers with several unique and extraordinary chemical structures and scaffolds with very potent biological activities. Marine invertebrates (animals, vascular plants, algae, cyanobacteria, fungi and actinomycetes) produce these unique and potent biologically active metabolites as a result of evolutional pressure in the form of predation, space and resources’ competition. The majority of these marine bioactive metabolites are of unique chemical structure and not reported in any terrestrial origin. It can be hypothesized that the more stressful the marine environment, the more unique and diverse metabolites are expected in marine organism (Seibel & Drazen, 2007). The Red Sea is only second to the great barrier reef in Australia in terms of the biodiversity of its fauna with versatile bioactive metabolites repertoire.

Rationale

In the recent couple of decades, intensive drug discovery research shed the light on the Red Sea marine invertebrates and its bioactive compositions. Several bioactive metabolites with unique chemical scaffolds were isolated, identified and showed promising pharmacological activities from the Red Sea marine life (El-Hossary et al., 2020). Biological activity herein is referred to in case of on any early experimental to late clinical evidence for potential use in the treatment of any human disease. On the other hand, the Persian Gulf and Gulf of Oman which are geographically close, atmospherically similar and under harsher environmental stress than the Red Sea, are surprisingly understudied in terms of bioactivity and drug discovery.

Relevance to readers

Herein we are utilizing the Red Sea as a benchmark marine environment and bioactive materials repertoire to provide a futuristic perspective for the high potential of the understudied and overlooked bioactive metabolites derived from the marine life of the Persian Gulf and Gulf of Oman despite its rich biodiversity under harsh environmental stress. This review would be of special interest for researcher interested in drug discovery and in particular, those who are interested in marine derived bioactive metabolites.

Survey Methodology

Search strategy

The search for all relevant studies published was conducted on PubMed and Google Scholar, with the search terms being “Marine metabolites; Bioactive metabolites; Marine bioactive metabolites; Drug discovery; Secondary metabolites; The Red Sea; The Persian Gulf; The Arabian Gulf; Gulf of Oman; Fauna; Aspergillus sp.”. Titles and abstracts were screened superficially for potential relevance, and once articles were deemed relevant, their full-texts were extracted and independently reviewed in depth to be considered for inclusion by the authors.

Inclusion criteria

Only Scopus-indexed journals were considered up to the year 2000. Eligible study designs included review articles, systematic reviews and meta-analyses. English was the sole publication language considered. Editorials, pre-prints that were not peer-reviewed, case reports, database reports were excluded.

Environmental Stress in the Persian Gulf and Gulf of Oman Compared to the Red Sea

The Persian Gulf/Gulf of Oman is a semi-enclosed sea, located in a subtropical region of the Middle East between latitudes of 24° and 30° North and longitudes 48° and 57° East. It is part of the Indo-Pacific Ocean ecoregion and consists of a shallow sedimentary basin with a relative depth of 35 m and covering a total area of 240,000 km2. The maximum depth of the Persian Gulf/Gulf of Oman is approximately 110 m and its photic zone extends between 6 to 15 m. The Iranian shores have steep sloping sections while the Persian shores are sedimentary with a gradual slope. Considering the combination of high latitudinal position, high evaporation rates, relative shallowness, coastline alterations, heavy traffic volume, frequent oil spills, excessive industrial effluents and restricted freshwater flow into the Gulf; the Persian Gulf/Gulf of Oman is subject to extreme environmental conditions. Therefore, marine organisms residing in the Persian Gulf/Gulf of Oman are surviving close to their environmental tolerance limits (Naser, 2014).

With respect to the Red Sea, it is also a semi-enclosed sea, between latitudes of 22° North and longitudes 38° East. It covers an area of approximately 438,000 km2 and is connected from its south end via the strait of Bab el Mandeb and Gulf of Aden to the Indian Ocean. The Red sea is known for its coastal fringing coral reefs and its clear and warm water. The central trough reaches over 2,000 m in the northern and central regions of the Red Sea. The sea bed rises from this trough to reach a depth of less than 300 to 400 m (Ellis et al., 2019).

The water temperature of the Persian Gulf fluctuates between 15 °C to 36 °C and the maximum recorded sea-surface temperatures was 38 °C in the summer of 1998 (Paparella et al., 2019). Whereas, the Red sea surface temperatures ranges from 18 °C to 31.5 °C. The temperature is lowest within the Gulf of Suez and increases gradually towards the southern half of the Red Sea (Gladstone, Facey & Hariri, 2006; Hariri, Gladstone & Facey, 2014).

The ecosystem in the Persian Gulf/Gulf of Oman is highly adapted to extreme environmental conditions but recent works have shown that it is not the maximum temperatures but rather the length of exposure that harms the marine life (Paparella et al., 2019). The reports demonstrated that mild bleaching occurs after a week of exposure to a sea-surface temperature of >35 °C and severe bleaching would occur after 3 weeks of exposure to temperatures ≥35 °C or even ≥34 °C exposure for 8 weeks. Therefore, it is observed that the average temperature-related stress in the Persian Gulf/Gulf of Oman is higher than in the Red Sea.

The Persian Gulf receives its freshwaters from Shatt al Arab waterway through three different rivers: Euphrates, Tigris and Karun. Although, recent damn constructions have resulted in significant reduction of freshwater entering the Persian Gulf through these rivers, together with waters from Hilleh, Mand and Hendijan rivers in Iran, the annual input of freshwater into the Gulf averages 110 km3 per year. On the other hand, the annual evaporation rates in the Persian Gulf ranges between 1.37 to 4.8 m. As a result, the marine system of Persian Gulf possesses high salinity >40 ppt, increasing to >50 ppt in the southern bays of open ocean conditions and exceeding 70 ppt in evaporative lagoons (Ben-Hasan & Christensen, 2019). Similarly, the evaporation rate in the Red sea reaches 1 to 2 m per year during both summer and winter. The Gulf of Aqaba experiences evaporation rate of 2 m per year and 2.35 m per year in the southern Red sea. As a result, the water salinity increases during summers reaching 37 ppt at Bab el Mandeb and 41 ppt towards the entrance of Gulf of Suez and Aqaba (Gladstone, Facey & Hariri, 2006; Hariri, Gladstone & Facey, 2014). The combined limited freshwater input with high evaporation rate makes the Red sea and similarly the Persian Gulf/Gulf of Oman to be the most saline water in direct contact with the world oceans (Naser, 2014; Ben-Hasan & Christensen, 2019).

The Persian Gulf and Gulf of Oman are known for their high shipping traffic either oil and gas import or foods and materials export. The high volume of shipping traffic and the sediments curbed within these ships ballast tank has resulted in the introduction of various exotic biotas to the marine system of the Persian Gulf/Gulf of Oman. The oil transportation alone accounts for 53,000 ships passing through the Hormuz corridor into the Persian Gulf annually. To a lesser extent, the Red Sea is navigated by some of the most prominent shipping lanes in the world. It is estimated that over 20,000 ships (oil and non-oil products) crosses through Bab el Mandeb strait annually (Sheppard, 2018). It was reported that several exotic zooplankton and phytoplankton organisms were collected from a ships ballast tank passing through the Persian Gulf (Al Muftah et al., 2016). It can be concluded that shipping traffic and its consequent marine life stress in the Persian Gulf significantly overweighs the Red sea.

The Persian Gulf is reported to have the largest oil reserves in the world and subsequently, oil spills and pollution in water constitutes a permanent stress and survival challenge to its marine life. The exploration, production and transportation of oil adds to the pollution in the Persian Gulf. Various sources of oil spills were reported in the Persian Gulf such as oil tanker incidents, offshore oil wells, weathered oil and tar balls, underwater pipelines, oil terminals, illegal ballast water dumping and military activities (Naser, 2014). The Persian Gulf/Gulf of Oman has witnessed several oil spill incidents in the near history, particularly in years 1980, 1990, 1991 and 2019 (Sale et al., 2011; Issa & Vempatti, 2018; Eidnes, Batalden & Sydnes, 2019). On the other hand, the Regional Organization for the Conservation of the Environment of the Red Sea and Gulf of Aden (PERSGA) prepared a partial report on Maritime incidents and accidents but did not provide any accurate record of spills. So far, there has been no report of a major oil spill or chemical pollution due to shipping incidents in the Red sea (Gladstone, Facey & Hariri, 2006). Yet, the number and the impact of oil spill incidents in the Persian Gulf/Gulf of Oman are way higher compared to the Red sea.

In addition to oil spills, the Persian Gulf is considered a hotspot for receiving intensive industrial effluents with high concentrations of hydrocarbons and heavy metals (Naser, 2014). The pollutants reside in the Persian Gulf for a considerable amount of time due to its relatively slow flushing time (between 3 to 5 years) and semi-enclosed nature. The organic pollutants from the agricultural runoffs enter the Persian Gulf through the rivers causing lower oxygen levels in coastal waters (Sale et al., 2011). Similarly, and due to the semi-enclosed nature of the Red sea, its water content renewal is limited. It is estimated that it would take 200 years for the renewal of the entire water in the Red sea. (Gladstone, Facey & Hariri, 2006; Hariri, Gladstone & Facey, 2014). Dumping toxic waste products in any semi-enclosed oceanic space such as the Persian Gulf or the Red Sea constitutes a considerable stress condition towards its marine life (Table 1).

Table 1 Comparative summary on general aspects of the Red Sea and Persian Gulf and Gulf of Oman.

Aspect	The Red Sea	The Persian Gulf & Gulf of Oman	
Temperature	Minimum sea-surface temperature: 18 °C

Maximum sea-surface temperature: 26.6 °C

	Sea temperature: 15 °C to 36 °C during the winter and summer seasons

Average sea-surface temperature: 37.7 °C

	
Salinity	The evaporation rate in the Red sea reaches 1 to 2 meters per year

	The annual input of freshwater into the Gulf averages 110 km3 per year

The annual evaporation rates ranges between 360 to 1,250 km3 per year

	
There is no significant difference in the salinity of the Persian Gulf and the Red sea except for some areas in the Persian Gulf where the salinity exceeds 40 ppt.

	
Shipping Traffic	Over 20,000 ships (oil and non-oil products) crosses through Bab el Mandeb strait annually

	The oil transportation accounts for 53,000 ships passing through the Hormuz into the Gulf annually

	
Shipping traffic in the Persian Gulf is much higher compared to the Red Sea.

There is no official data available on the overall shipping traffic of the Persian Gulf as just the oil shipment in the Persian Gulf is two times more than the overall shipping traffic in the Red sea.

	
Oil Spill	The Persian Gulf has witnessed significant oil spill incidents in the world

The number of oil spills in the Red sea has not been recorded

It is evident that the number of oil spills in the Persian Gulf is way higher when comparing it to the Red sea

	
Industrial Effluents	An estimate of 200 years for the renewal of the entire water

	The flushing time of the seawater is between 3 to 5 years

	

Common Microorganism Isolates in the Persian Gulf and Gulf of Oman in Relation to the Red Sea

Due to the environmental stress conditions in marine life, many organisms pull through by releasing secondary metabolites which might be toxic and bioactive at the same time (Vaseghi et al., 2018). These bioactive compounds may be considered as an important source for drug discovery. Soft corals, seaweed and algae, sponges, bacteria, and fungi are some of the marine organisms that have been researched for potential bioactivity (Dobretsov et al., 2016). They have been reported to elaborate alkaloids, anthraquinones, ethers and many other chemical families that possess antioxidant, antimicrobial, anti-inflammatory, antitumor and cytotoxic activities (Lee et al., 2013a).

Fungi are more environment friendly to study than other marine corals and organisms. They are easily isolated and grown in the lab which makes it safer for the marine environment. The extreme conditions of marine life have caused fungi to become stress-resistant by forming many useful secondary metabolites (Alwakeel, 2017). These metabolites are said to have important biological activities including antimicrobial, anticancer, antioxidants and even antidiabetic, which is one of the reasons fungi are becoming more popular in marine drug discovery research (Hasan et al., 2015).

The Red Sea and Persian Gulf and Gulf of Oman are rich in marine fungi, and therefore, rich in bioactive compounds due to their high stress environments enhancing the synthesis and release of toxic and bioactive chemicals. Several common organisms have been found in both the Red Sea and Persian Gulf such as, Aspergillus sp., Penicillium sp., Talaromyces sp. and Trichoderma sp. (Nosratabadi et al., 2017).

Nonetheless, Aspergillus species have been found and extensively researched in the Red Sea marine environment. However, Aspergillus species isolated from the Persian Gulf such as, A. flavus, A. niger, A. fumigatus and A. terreus are significantly under-researched compared to the same species isolated from the Red Sea. In addition, many other Aspergillus species were found in the Red Sea (A. sydowii, A. ochraceus, A. unguis, A. clavatus, A. versicolor, A. utus and A. caespitosus) but yet to be isolated and reported from the Persian Gulf (Alwakeel, 2017). This leads us to the following questions: (1) Are these fungi unable to survive the marine stress of the Persian Gulf rather than surviving the environmental stress of the Red Sea or simply not isolated yet? (2) Are the same Aspergillus species isolated from the Persian Gulf likely to synthesize the same portfolio of bioactive compounds like those isolated from the Red Sea?

Aspergillus is considered a terrestrial fungus which can survive in the marine environment in specific conditions. It has numerous species that produce multiple bioactive metabolites. Bioactive metabolites from A. fumigatus show potent broad spectrum antimicrobial (Shaaban et al., 2013). A. versicolor has plenty of bioactive compounds which were found to be antioxidant, antibacterial, fungicidal, and lipid lowering agents (Ahmed et al., 2017). A. unguis possesses a bioactive metabolite with α-glucosidose inhibitory, antimicrobial and antioxidant effects (Abd El-Hady et al., 2015). Indole alkaloids isolated from A. ochraceus shows antitumor effects against several tumor types. A. utus also possesses cytotoxic metabolites against several cancer types (Lee et al., 2013b).

Biological Activities and Chemical Structures of Some Secondary Metabolites Isolated from Aspergillus spp.

The fungal flora, Aspergillus in specific, is understudied in the Persian Gulf. However, according to the available literature, A. flavus, A. fumigatus, A. niger and A. terreus were found in the soil sediment of the Persian Gulf. It is worth mentioning that A. fumigatus, A. flavus, and A. niger account for more than 95% of the Aspergillus species pathogens (Nosratabadi et al., 2017).

Aspergillus flavus is a saprophytic soil type fungus which contaminates and infects seed crops. It is best known for aflatoxin secondary metabolite, which acts as a potent carcinogenic compound (Cary et al., 2018) and aspergillosis disease which affects immunocompromised patients (Amaike & Keller, 2011; Cary et al., 2018). The aflatoxins are highly substituted difuranocoumarins fused with dihydrofurofuran moiety (Marchese et al., 2018). Different toxins belong to aflatoxins family such as Aflatoxin B1 (AFB1(1)) which is known to be the most harmful and Aflatoxin M1 (AFM1(2)) (Table 2), the hydroxylated metabolite of AFB1 (1), both are classified as human carcinogens through the activation of CYP450 (Marchese et al., 2018).

Table 2 Example of some bioactive metabolites derived from the marine fungi Aspergillus flavus.

Aspergillus flavus	

Aflatoxin B1 (AFB1) (1)	
Aflatoxin M1 (AFM1) (2)	
Aflatoxin B2a (3)	

Aspergillus fumigatus belongs to the filamentous fungi family that is present in the environment and can cause many diseases and life-threatening conditions in immunocompromised patients (Van De Veerdonk et al., 2017). Due to the stressful environmental conditions, A. fumigatus adapts via the production of several secondary metabolites and different mycotoxins such as, gliotoxin (4), fumagillin (5) and pseurotin A (6) (Table 3). Interestingly, A. fumigatus is reported to produce up to 226 secondary metabolites. Since the time of its discovery; studies focused on the biological activities of fumagillin such as its antitumor, antibacterial and antiparasitic effects. (Guruceaga et al., 2020).

Table 3 Example of some bioactive metabolites derived from the marine fungi Aspergillus fumigatus.

Aspergillus fumigatus	

Gliotoxin (4)	
Fumagillin (5)	
pseurotin A (6)	

Aspergillus niger is a human pathogenic filamentous ascomycete fungi which is known to have a vital economic role in the industrial scale fermentation for the production of citric acid (Baker, 2006). In addition, A. niger is known to produce ochratoxin A (7) and fumonisins (8–13) (Table 4). Fumonisins are toxic and carcinogenic metabolites, amongest Fumonisins B1 (FB1) (8), B2 (FB2) (9) and B3 (FB3) (10) are the most abundant in the fungal cultures (Baker, 2006; Tamura et al., 2014).

Table 4 Example of some bioactive metabolites derived from the marine fungi Aspergillus niger.

Aspergillus niger	

Ochratoxins A (7)	
Fumonisins (8-13)	

Aspergillus terreus is an important bio-processor fungus as it is useful in the production of many bioactive metabolites such as, itaconic acid (14) and metatartaric acid (15) lovastatin (16), territrem B (17) and terreulactone (18) (Table 5). In addition, there are many other mycotoxins secondary metabolites of A. terreus such as, patulin (19) (Table 5), ochratoxins A (7) (Table 4), citrinin (20), emodin (21) and sulochrin (22) (Table 5) (Varga et al., 2005). Amongst the above, lovastatin (16) is the first to be discovered (HMG)-CoA reductase inhibitor with potent antihyperlipidemic effects. Another notable metabolite of A. terreus is Terrein (23) (Table 5) that was isolated for the first time in 1935 and attracted a lot of attention due to its versatile bioactivities including inhibition of plant growth, anti-microbial, anti-proliferative, and anti-oxidative activities (Zhang et al., 2018). Yet, the chemical structures and accordingly the biological activities of A. terreus isolated secondary metabolites are non-surprisingly very versatile (Boruta & Bizukojc, 2017).

Table 5 Example of some bioactive metabolites derived from the marine fungi Aspergillus terreus.

Aspergillus terreus	

Itaconic acid (14)	
Metatartaric acid (15)	
Lovastatin (16)	

Territrem B (17)	
Terreulactone (18)	
Patulin (19)	
Ochratoxins A (7)	

Citrinin (20)	
Emodin (21)	
Sulochrin (22)	
Terrein (23)	

Asperigllus versicolour is a marine endophytic fungus that is isolated from the inner tissue of the green alga called Halimeda opuntia found in the Red Sea (Finefield et al., 2011). A. versicolour contain diverse set of secondary metabolites such as Diketopiperazine (DKP) (24), Fellutamides F (25), sterigmatocystin (26) , Dihydrosterigmatocystin (27), Cottelosines A and B (33 and 34) several anthraquinones (28-32), isorhodoptilometrin-1-methyl ether (35), siderin (37), emodin (22) evariquinone (36), arugosin-C (38), and variculanol (39) (Table 6), sesquiterpenoid nitrobenzoyl esters, chormone derivatives, meroterprnoid and quinazolinone alkaloids which are reported to possess anti-cancer, antibacterial, insecticidal, fungicidal, antioxidant, lipid lowering properties and HCV protease inhibitory action (Lee et al., 2010, 2011; Zhuang et al., 2011; Hawas, El-Beih & El-Halawany, 2012; Ahmed et al., 2017)

Table 6 Example of some bioactive metabolites derived from the marine fungi Aspergillus versicolor.

Aspergillus versicolor	

Diketopiperazine (DKP) (24)	
Fellutamides F (25)	
Sterigmatocystin (26)	

Dihydrosterigmatocystin (27)	
Anthraquinones (28–30)

(31–32)
R=O
R=OH	

Cottelosines A and B (33 and 34)	
Isorhodoptilometrin-1-methyl ether (35)	
Evariquinone (36)	

Siderin (37)	
Arugosin C (38)	
Variculanol (39)	

Aspergillus ochraceus is a marine mitosporic fungus from the trichocomaceae family which was isolated from brown mold/clay of the Red Sea sediments. It can also be found in dried food, such as; dried beans, nuts dried fish and oilseeds (de W Blackburn, 2006). Several secondary metabolites were isolated from A. ochraceus such as the phenylated indole alkaloids Stephacidines (40,41) (Table 7) which showed potent antitumor activity against several tumor types (Artman, Grubbs & Williams, 2007; Finefield et al., 2011; Lee et al., 2013b). Yet, Stephacidine B (41) is the most complex isolated alkaloids found until the beginning of the twenty first century (Zhuang et al., 2002).

Table 7 Example of some bioactive metabolites derived from the marine fungi Aspergillus ochraceus.

Aspergillus ochraceus	

Stephacidines A (40)	
Stephacidines B (41)	

Conclusion

The marine environmental stress conditions indue many faunae and symbiont microorganisms to synthesize and release secondary metabolites of unique structure and interesting biological activities. These bioactive compounds can serve as an important source for drug discovery. The Persian Gulf/Gulf of Oman have a significantly harsher environment when compared to the Red Sea due to fluctuating temperatures between summer and winter which leads to high evaporation rates. Moreover, high salinity also contributes to the harsh environment leading microorganisms to live close to their limits of environmental tolerance. The Red Sea marine symbionts derived bioactive metabolites are very well studied while very much untapped in the Persian Gulf/Gulf of Oman. In the current perspective, and due to similar fungal species isolated from both environments, we are expecting interesting and unique secondary metabolites to be found and isolated from the marine life of the Persian Gulf/Gulf of Oman. The authors believe that using the Red Sea as a reference to target similar microorganisms that have elaborated bioactive metabolites is worthwhile. With the extreme environmental conditions in the Persian Gulf and Gulf of Oman, it is likely that marine research will lead to the isolation and identification of novel metabolites of medicinal value.

Additional Information and Declarations

Competing Interests

Author Contributions

Data Availability

The authors declare that they have no competing interests.

Reham K. Abuhijjleh performed the experiments, prepared figures and/or tables, authored or reviewed drafts of the paper, and approved the final draft.

Samiullah Shabbir performed the experiments, prepared figures and/or tables, authored or reviewed drafts of the paper, and approved the final draft.

Ahmed M. Al-Abd conceived and designed the experiments, analyzed the data, authored or reviewed drafts of the paper, and approved the final draft.

Nada H. Jiaan performed the experiments, authored or reviewed drafts of the paper, and approved the final draft.

Shahad Alshamil performed the experiments, authored or reviewed drafts of the paper, and approved the final draft.

Eman M. El-labbad performed the experiments, analyzed the data, prepared figures and/or tables, authored or reviewed drafts of the paper, and approved the final draft.

Sherief I. Khalifa conceived and designed the experiments, analyzed the data, authored or reviewed drafts of the paper, and approved the final draft.

The following information was supplied regarding data availability:

This is a review article; there is no raw data.

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
