# Peer review of "Bioactive marine metabolites derived from the Persian Gulf compared to the Red Sea: similar environments and wide gap in drug discovery"

_PeerJ, doi:10.7717/peerj.11778_

## Round 0.1 · original submission · Major Revisions

As you can see in the reviewer's reports, one of them didn't recommend the publication of the manuscript, while the other three are favorable, once you make major revisions.

Please, give special attention to the comments of reviewers #3 and #4.

A new version of your manuscript will be reevaluated by me and the reviewers.

Reviewer 1 ·

Basic reporting

No comment

Experimental design

No comment

Validity of the findings

The article in this paper is a review, so it has no findings.

Additional comments

The article is well written however, some doubts have arisen on the subject addressed in the paper:
1- The author states, in the introduction, that marine metabolites are unique and that the maritime environment interferes with the characteristics of these metabolites. He also reported that the marine environment of the gulfs is more stressful than the Red Sea. How was it possible to compare metabolites with so many differences?
2 - What should be in the structure to be considered a compound with potential for treatments?
3- What are the characteristics of the metabolites found in the gulfs that make them equally or more important than the metabolites found in the Red Sea?

Reviewer 2 ·

Basic reporting

no comment

Experimental design

no comment

Validity of the findings

no comment

Additional comments

The manuscript entitled "Bioactive marine metabolites derived from The Persian Gulf compared to The Red Sea: Lessons in drug discovery" is an attempt to deliver a comparative discussion of the bioactive marine metabolites present in different environments (the Persian Gulf and the Red Sea). However, authors have spent a long part of the text comparing unnecessary technical information, instead of focusing on its main goal. Furthermore, when the main point of the manuscript was reached (Isolated microorganism and Bioactive chemical structures) the text was brief and shallow, not presenting an appropriate discussion and comparison, completely leaving the focus proposed in the title of the work, that would be a lesson in drug discovery. Thus, in my opinion, the manuscript does not deserve to be accepted for publication in PeerJ.

Reviewer 3 ·

Basic reporting

About table 1:

First of all, it is call a table, however very confusing. Tables should have entries, so I suggest the authors to structure better the table.

Structures Ochratoxins A (7) and Fumonisins (8-13) could change places, so I suggest Ochratoxins A (7) in the left and Fumonisins (8-13) in the right.

In the Aspergillus versicolor, its written Anthraquinones (28-31), however there are drawn componds 28, 29 and 30, there is no 31. At the same entry, the next cell shows compounds 30 and 31, however compounds 30 from Anthraquinones (28-31) are not the same.

I suggest to redraw compounds to 39 e 40.

Experimental design

no comment

Validity of the findings

The authors states interesting questions about fungi species, making a comparition of Red Sea and Persian Gulf and Gulf of Oman. Using some general aspects of each class of Aspergilium (well supported by literature), authors try to stimulated curiosity and open the eyes of researcher to new possibilities.

Additional comments

In general, the review can encourage other scientists and research centers. To make the reader more comfortable, I suggest that the authors prepare a comparative table on general aspects of the Red Sea and Persian Gulf and Gulf of Oman. As well as for the species of Aspergilium observed in each of the seas.

I recommend this manuscrips for publication after revision.

Reviewer 4 ·

Basic reporting

The article deals with a review comparing marine bioactive metabolites in the Persian Gulf with the Red Sea, produced by fungi of the Aspergillus species.

Experimental design

The methodology is consistent, and the sources are adequately cited in the paper.

It is suggested a revision in the name of the mentioned metabolites, sometimes with a capital letter and sometimes with a small letter.

Validity of the findings

Although it is a short review (mine review), the results are interesting, and the work well structured.

Additional comments

This revision work needs minor corrections, such as those mentioned above. A revision in the name of the metabolites and a complementation of the literature.

---

## Round 0.2 · accepted · Accept

The reviewers and I have made an evaluation of the new version of the review and we believe it can be accepted for publication in PeerJ.

Reviewer 3 ·

Basic reporting

no comment

Experimental design

no comment

Validity of the findings

no comment

Additional comments

In general, the authors have answered all questions and performed all modifications that were previously made to the manuscript by all reviewers.

So, in my opinion, once all points have been covered, I recommend this article for publication on PeerJ